# Structure of the bacterial flagellar hook cap provides insights into a hook assembly mechanism

Hideyuki Matsunami [1,2,6 ✉], Young-Ho Yoon[1], Katsumi Imada [3], Keiichi Namba [2,4 ✉] & Fadel A. Samatey [1,5,7 ✉]

Assembly of bacterial flagellar hook requires FlgD, a protein known to form the hook cap. Symmetry mismatch between the hook and the hook cap is believed to drive efficient assembly of the hook in a way similar to the filament cap helping filament assembly. However, the hook cap dependent mechanism of hook assembly has remained poorly understood. Here, we report the crystal structure of the hook cap composed of five subunits of FlgD from *Salmonella enterica* at 3.3 Å resolution. The pentameric structure of the hook cap is divided into two parts: a stalk region composed of five N-terminal domains; and a petal region containing five C-terminal domains. Biochemical and genetic analyses show that the N-terminal domains of the hook cap is essential for the hook-capping function, and the structure now clearly reveals why. A plausible hook assembly mechanism promoted by the hook cap is proposed based on the structure.

[1] Trans-Membrane Trafficking Unit, Okinawa Institute of Science and Technology Graduate University, 1919-1 Tancha, Onna, Kunigami, Okinawa 904-0495, Japan. [2] Graduate School of Frontier Biosciences, Osaka University, 1-3 Yamadaoka, Suita, Osaka 565-0871, Japan. [3] Department of Macromolecular Sciences, Graduate School of Sciences, Osaka University, 1-1 Machikaneyama-cho, Toyonaka, Osaka 560-0043, Japan. [4] RIKEN Spring-8 Center and Center for Biosystems Dynamics Research, 1-3 Yamadaoka, Suita, Osaka 565-0871, Japan. [5] Pineuretics LLC, 1686 Shimabuku, Kitanakagusuku, Nakagami, Okinawa 901-2301, Japan. [6] Present address: Molecular Cryo-Electron Microscopy Unit, Okinawa Institute of Science and Technology Graduate University, 1919-1 Tancha, Onna, Kunigami, Okinawa 904-0495, Japan. [7] Present address: Microbial Sciences Institute, Department of Molecular Biophysics and Biochemistry, Yale University, New Haven, CT 06511, USA. ✉email: hideyuki.matsunami@oist.jp; keiichi@fbs.osaka-u.ac.jp; fadel.samatey@yale.edu

The bacterial flagellum is a locomotive organelle for cellular motility[1]. Many bacteria use flagella for motility coupling to chemotaxis[2]. The bacterial flagellum is a macromolecular nanomachine composed of a filament as a helical proper, a hook as a universal joint, and a basal body containing a driving shaft and a rotary motor[3]. In the bacterial flagellum, capping proteins FlgJ, FlgD, and FliD are essential for the assembly of axial proteins by binding to the growing tips of the rod, the hook, and the filament, respectively[4–6]. These cap proteins share no overall primary sequence similarity to one another and differ in their functions during the flagellar formation.

In *Salmonella enterica* serovar Typhimurium, the flagellar assembly mechanism has been intensively investigated[7,8]. FlgJ, the rod cap protein, has an enzymatic function of muramidase in its C-terminal domain to digest the peptidoglycan (PG) layer locally around the rod in bacteria cells. The N-terminal domain of FlgJ is believed to bind to the growing tip of the rod as a capping protein and help assembly of the rod component proteins (FlgB, FlgC, FlgF, and FlgG) to form the rod through the PG layer[9]. For the rod to penetrate the PG structure of cells, the muramidase domain of FlgJ must digest the PG just around the rod by anchoring to the tip of the growing rod through the N-terminal domain of FlgJ[10]. Although crystal structures of the C-terminal domain of FlgJ have been solved[11,12], neither the structure of the entire FlgJ nor the quaternary structure of FlgJ cap is available.

FliD, also called HAP2 (hook-associated protein 2), is known to cap the filament by forming a pentamer at the tip of the growing filament. HAP2 prevents the filament protein FliC exported to the filament tip from diffusing away and helps its folding during incorporation into the filament[13,14]. In solution, HAP2 shows diverse oligomerization states such as monomeric, pentameric, and decameric[15]. A detailed structure of HAP2 from *S. enterica* revealed by electron cryomicroscopy (cryoEM) with single-particle analysis showed that HAP2 has a pentagonal base plate that looks like a lid and five leg domains associating to protofilaments at the distal end of the filament[16,17]. A rotational model of the HAP2 cap at the filament tip for filament assembly was also inferred from the pentameric structure bound to the 11 protofilament structure of the filament with a symmetry mismatch[16,17]. Recently, it has been reported that the central core fragments of HAP2 from *Pseudomonas aeruginosa*, *Escherichia coli*, *S. enterica*, and *Serratia marcescens* form a hexamer, a hexamer, a pentamer, and a tetramer, respectively, indicating that FliD is structurally versatile in terms of its oligomerization state from species to species[18–20].

The flagellar hook cap protein FlgD is essential for hook assembly[4,21]. In the absence of FlgD, the hook protein FlgE is secreted out of the cell without polymerizing into the hook. *S. enterica* with a *flgD*-deficient genetic background shows no motility because of the failure of hook and filament assembly. In the presence of the hook cap, FlgE molecules that are exported to the growing end of the hook through the central channel of the rod and hook get folded and incorporated into the hook until the hook length reaches around 55 nm[22]. The N-terminal 86 amino acid residues of FlgD from *S. enterica*, which is 232 residues in full length, can complement the flagellation of a FlgD-null mutant strain to some extent[23]. This FlgD fragment is likely to form a minimum cap complex that somehow exerts its function during hook assembly. Pseudorevertants with mutations within the *flgD* gene have been isolated from hook assembly-deficient *flgE* mutants in *S. enterica*[24]. Recently, structures of FlgD fragments from *Xanthomonas campestris* pv. campestris[25], *P. aeruginosa*[26], and *Helicobacter pylori*[27] were solved by X-ray crystallography. These crystal structures, however, exclusively contain the C-terminal domains. The complete structure of FlgD as the hook cap remain elusive because of the difficulty in crystallizing full-length FlgD proteins.

Although the structure of the hook from *S. enterica* and the dynamic mechanism of the hook function as a molecular universal joint have been reported[28–31], no cap-bound structure of the hook has ever been reported and this has hampered our understanding of the hook assembly mechanism driven by the capping protein. We hereby describe the first pentameric structure of the flagellar hook cap revealed by X-ray crystallography and provide a possible hook assembly mechanism with an aid of the hook cap.

## Results

**Characterization of the hook cap from *S. enterica*.** Strains and plasmids used in this study are summarized in Supplementary Table 1. *S. enterica* FlgD (232 amino acid residues, hereafter referred to as FlgD) was overproduced in *E. coli* BL21(DE3)pLysS without any affinity tags and purified by three steps of column chromatography. In the last step of the purification, fractions containing an oligomer and monomer of FlgD were separated by gel-filtration chromatography (Fig. 1a, b). Blue-native gel electrophoresis showed the oligomer and monomer migrating as molecular sizes of ~200 and 40 kDa, respectively (Fig. 1c). These rates of retardation of the bands reflected the molecular shapes of these species being far from general globular forms, just as found in the gel-filtration profile. Chemical cross-linking experiment showed that FlgD formed predominantly pentamer and decamer, which is probably composed of two pentamers (Fig. 1d, e). Limited proteolysis using trypsin revealed that the N-terminal domain of FlgD was more susceptible for digestion in the monomeric form than in the pentamer (Supplementary Fig. 1a, b). In the monomer, three major fragments were identified as Δ16 (residues from Thr 17 to the C-terminal Ile232 of FlgD), Δ45 (from Asn 46), and Δ73 (from Leu-74) by time-of-flight mass spectroscopy. These digestion sites were all located in the N-terminal domain of FlgD, suggesting that the N-terminal regions of FlgD is involved in the pentamer formation.

**Crystal structure of the hook cap.** The crystal structure of the hook cap was determined at 3.3 Å resolution (Fig. 2a). The data collection and refinement statistics are summarized in Table 1 and Supplementary Table 2. Five subunits of FlgD forming the pentameric complex are found in the asymmetric unit of the crystal. The shape of the pentamer looks like a flower with a petal-shaped head and a stalk region (Fig. 2a and Supplementary Movie 1). In the crystal, most residues of FlgD are well-ordered, except for the very N-terminal region of each subunit. The N-terminal region containing the N terminus of each subunit, with lengths ranging from 55 to 65 residues, could not be modeled due to their poor electron densities, although the N-terminal region spanning from residues 29 to 45 were predicted to form an α-helix by *PSIpred*[32] (Supplementary Fig. 2a). Out of 232 residues of full-length FlgD, the final model contains residues 56–232 in subunit A, 62–232 (except for 113–118) in B, 63–232 (except for 207–214) in C, 66–232 (except for 113–118, 131–132, and 165–167) in D, and 58–232 in E (Supplementary Fig. 2b). When one FlgD molecule is viewed as in Fig. 2b, which corresponds to subunit A colored green in the right panel of Fig. 2a, FlgD can be divided into two distinct domains: the long N-terminal α-helix (α1; Asn56–Gly98) forming a five-stranded helix bundle of the stalk and a β-structure-rich domain formed by the remaining C-terminal chain (β1–β12), previously defined as a hybrid structure of a tudor and a fibronectin type III domain[23] (Fig. 2b). Prior to solving the crystal structure of the hook cap, the crystal structure of a C-terminal fragment of FlgD (FlgD$_{74-232}$) was determined at a resolution of 2.2 Å (Fig. 2c). The data collection and refinement statistics are summarized in Table 2 and

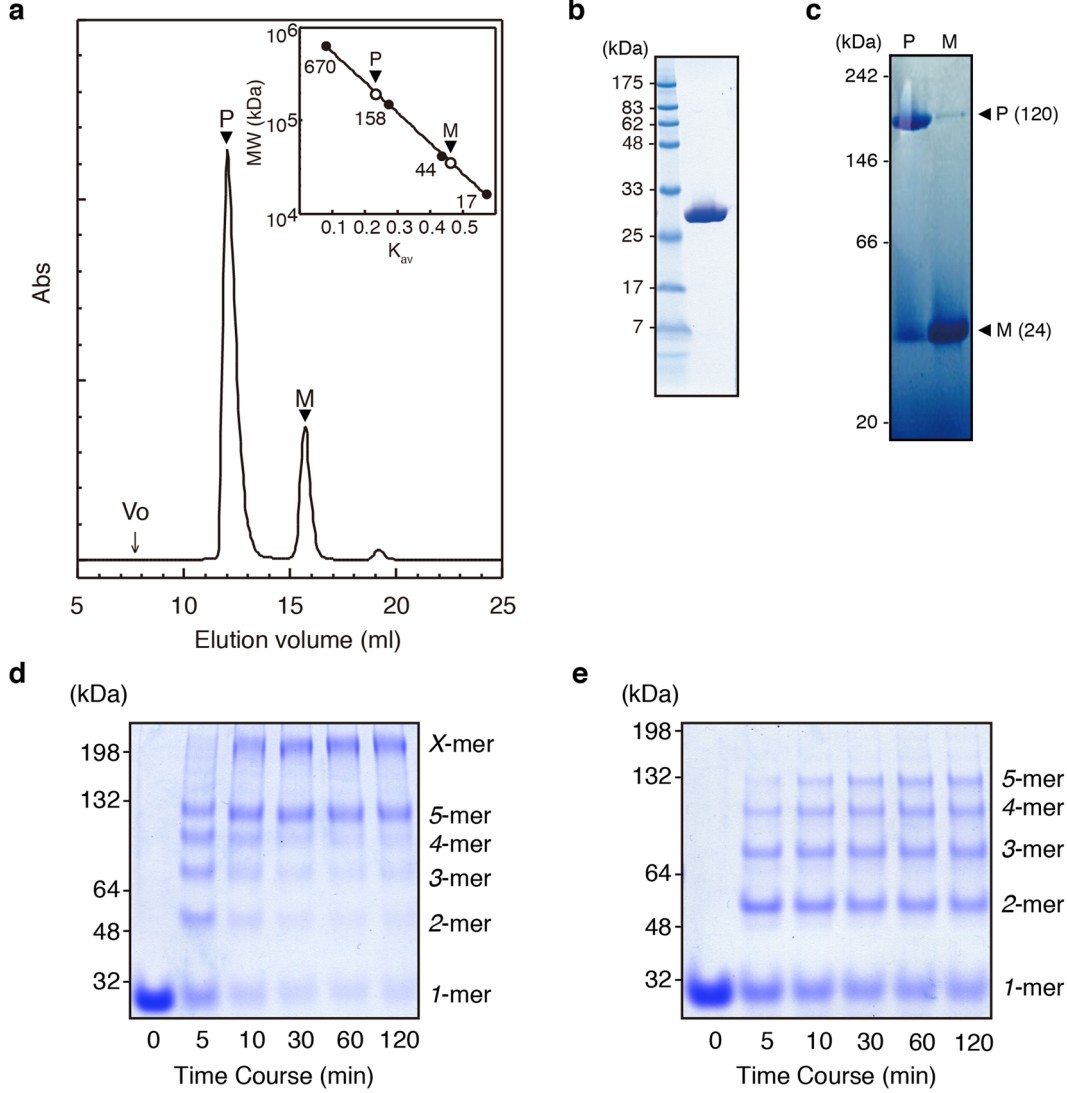

**Fig. 1 Biochemical characterization of *S. enterica* FlgD. a** A typical gel-filtration profile of purified FlgD. A FlgD solution was applied on Superdex 200 HR (GE Healthcare) equilibrated with 20 mM sodium phosphate (pH 6.8) containing 150 mM sodium chloride (Vo, void volume; P, pentamer; M, monomer). Inset: estimation of the molecular masses of FlgD pentamer (P) and monomer (M). The molecular size of standard proteins (kDa) was indicated on the left. **b** SDS-PAGE analysis of purified FlgD used for gel-filtration analysis. Standard molecular masses (kDa) are indicated on the left. **c** BN-PAGE analysis of the fractions containing the FlgD pentamer or monomer. Standard molecular masses (kDa) are indicated on the left. Chemical cross-linking analyses of purified FlgD. **d**, **e** SDS-PAGE of cross-linked products. Products by EDC with Sulfo-NHS at a molar ratio of 1 : 2000 (**d**), in which a product with a higher molecular weight is indicated as "*X*-mer," and products by DTSSP at a molar ratio of 1 : 50 (**e**). The molecular weight markers of cross-linked hemoglobin (32, 48 and 64 kDa) and albumin (132 and 198 kDa) were indicated on the left.

Supplementary Table 3. Further structural details of FlgD$_{74-232}$ is described afterwards. The atomic model of the five subunits of the hook cap could be superimposed well through the C-terminal domains, with a root-mean-square (rms) differences of Cα atoms <1.5 Å, whereas the N-terminal domains were flexible enough to form multiple conformations in the crystal structure (Fig. 2d).

**Comparison with the structure of a fragment FlgD$_{74-232}$.** The crystal structure of FlgD$_{74-232}$ contained a dimer in the asymmetric unit and the models of the two molecules were built for residues Ser88–Ile232 (except for 188–194 and 208–212) in chain A and Ser88–Ile232 (except for 116–119 and 188–192) in chain B (Supplementary Fig. 2c), which cover the last two turns of the N-terminal α-helix and the entire C-terminal β-structure-rich domain of FlgD in the pentamer. The rest of the fragment including the poly-histidine tag attached for fragment purification

was not built due to poor electron density. The structures of the two molecules in the asymmetric unit were nearly identical to each other with an rms difference of their Cα atoms of 0.7 Å. The structures of the fragments were also nearly the same as those of FlgD subunits in the pentamer, except for the N-terminal region removed prior to crystallization. The rms differences of the Cα atoms between them ranged from 0.5 to 1.2 Å. Comparison with previously solved FlgD structures deposited in the Protein Data Bank indicates that the C-terminal domain shares a conserved structure (Supplementary Fig. 3a–d), although the sequence similarities between FlgD proteins are not so high (Supplementary Fig. 3e).

Interestingly, we found a molecular interaction in the crystal packing of FlgD$_{74-232}$ that is closely related to the one for pentamer formation. The interface between the two neighboring dimers in the crystal was nearly identical to that of the neighboring subunits in the pentamer model (Supplementary

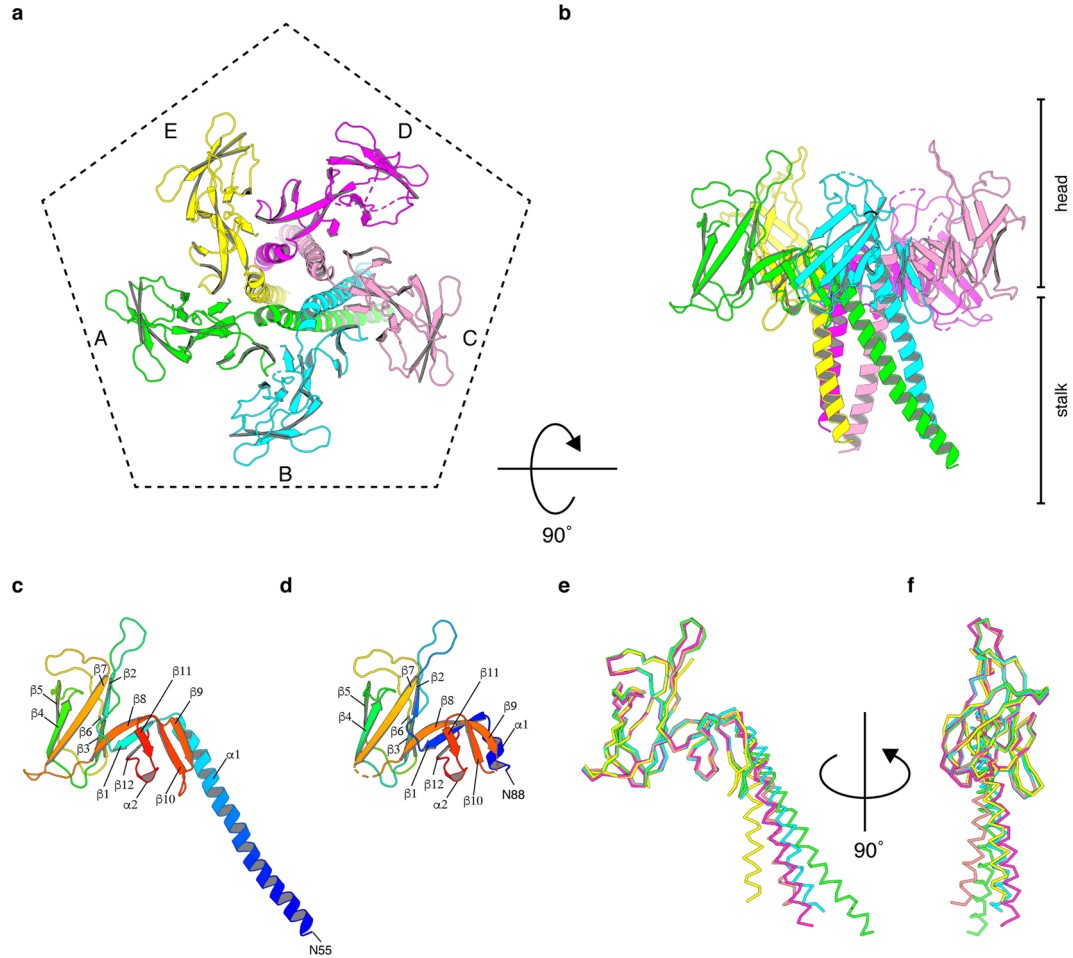

**Fig. 2 Crystal structures of the hook cap from *S. enterica*. a, b** A Cα ribbon representation of the hook cap colored by subunits (A, green; B, cyan; C, pink; D, magenta; E, yellow), viewed from the top (**a**) and side (**b**) of the hook cap. **c** The subunit A of the hook cap in **a** and the chain A of FlgD$_{74-232}$ (**d**) depicted in rainbow color and labeled with secondary structures. **e, f** Superimposition of the five subunits of the hook cap (chain A–E) in **a** for comparison viewing from two directions.

### Table 1 Structural refinement statistics of *S. enterica* FlgD.

|  | *S. enterica* FlgD Native |
| --- | --- |
| Space group | $P3_221$ |
| Unit-cell parameters (Å) | $a = b = 141.28$, $c = 153.49$ |
| Number of molecules in the asymmetric unit | 5 |
| Resolution (Å) | 47.2–3.3 (3.418–3.3 |
| $R_{work}$ | 0.2465 (0.2803) |
| $R_{free}$ | 0.2875 (0.3199) |
| Number of non-hydrogen atoms |  |
| Protein | 5988 |
| Water | - |
| B-factors |  |
| Protein | 100.7 |
| Water | - |
| RMS |  |
| Bond length (Å) | 0.003 |
| Bond angles (°) | 0.79 |
| Ramachandran plot (%) |  |
| Favored | 99.0 |
| Allowed | 1.0 |
| Outliers | 0.0 |

Statistics for the highest-resolution shell are shown in parentheses.

Fig. 4a) and, therefore, at least for the C-terminal domains, the pentamer model of the hook cap could be built by putting five subunits using this molecular interface (Supplementary Fig. 4b).

**Inter-subunit interaction in the hook cap.** Polar contacts found in the crystal structure of the hook cap are summarized in Fig. 3 and Supplementary Table 4. In the N-terminal domain, there are significant polar contacts between residues Ser-91 of subunit A and Gln-92 of subunit B, as well as Gln-92 of subunit A and Ser-91 of subunit E (Fig. 3a, c). In the loop region of subunit A between β9 and β10 (see Fig. 2c), the polar contacts are also formed with the C-terminal region of subunit B, as found in the C-terminal region of subunit A with the loop region of subunit E (Fig. 3b, d).

**Model of the hook cap on the distal end of the hook.** The structure of the hook from *S. enterica* was determined by cryoEM by Fujii et al.[31]. Although the hook diameter is 180 Å, the diameter of the hook cap was estimated from the crystal structure to be ~90 Å. When the hook cap model was docked on to the distal tip of the hook model (with 11 FlgE molecules for clarity), most part of the hook cap was encapsulated within the large central cavity formed at the tip of the hook and lay over the D0 and D1 domains of the hook protein FlgE (Fig. 4a, b). The interactions between the stalk of the hook cap with the central channel of the

**Table 2 Summary of the refinement statistics for the *S. enterica* FlgD$_{74-232}$ fragment.**

|  | *S. enterica* FlgD$_{74-232}$ Native |
|---|---|
| Space group | $P2_12_12$ |
| Unit-cell parameters (Å) | $a = 75.99$, $b = 104.32$, $c = 43.87$ |
| Number of molecule in the asymmetric unit | 2 |
| Resolution (Å) | 33.57–2.2 (2.260–2.2) |
| $R_{work}$ | 0.1999 (0.2460) |
| $R_{free}$ | 0.2481 (0.3949) |
| Number of non-hydrogen atoms |  |
| Protein | 1967 |
| Water | 117 |
| B-factors |  |
| Protein | 58.9 |
| Water | 59.1 |
| RMS |  |
| Bond length (Å) | 0.007 |
| Bond angles (°) | 1.03 |
| Ramachandran plot (%) |  |
| Favored | 99.0 |
| Allowed | 1.0 |
| Outliers | 0.0 |

Statistics for the highest-resolution shell are shown in parentheses.

hook formed by the D0 domains of FlgE are not clear yet without the complete N-terminal structure of FlgD in the hook cap. However, as the overall shape of the hook cap is quite distinct from that of the filament cap, the hook and filament caps would adopt different structural mechanisms for facilitating the folding and assembly of unstructured FlgE and FliC proteins after their translocation through the narrow central channel of the hook and filament, respectively[16].

**Mutational analysis of the N-terminal domain of FlgD.** Although the N-terminal domain of FlgD is known to be essential for the hook cap function in hook assembly, the roles of its individual small parts remained unknown. To investigate which parts of the N-terminal domain are important for the hook cap function, we carried out a ten-amino-acid deletion analysis of FlgD. We constructed a series of FlgD variants from Δ1 to Δ10, each with deletion of ten amino acid residues (Fig. 5a), and analyzed their complementation effects on the motility of a *flgD* mutant stain, SJW156, on a soft-agar plate (Fig. 5b). Wild-type FlgD restored the motility of the *flgD* mutant. Cells transformed with *flgD* variant Δ2 showed a weakly motile phenotype and those with Δ3 showed a swarming circle, albeit much smaller than the wild type. Cells transformed with Δ4 to Δ9 were non-motile. Cells with Δ10 showed a swarming size nearly the same as the wild type, probably because almost the entire N-terminal domain is intact as it is for the wild type. These results clearly indicate that the N-terminal domain is actually very important for the hook cap function. Expression and secretion of these FlgD variants

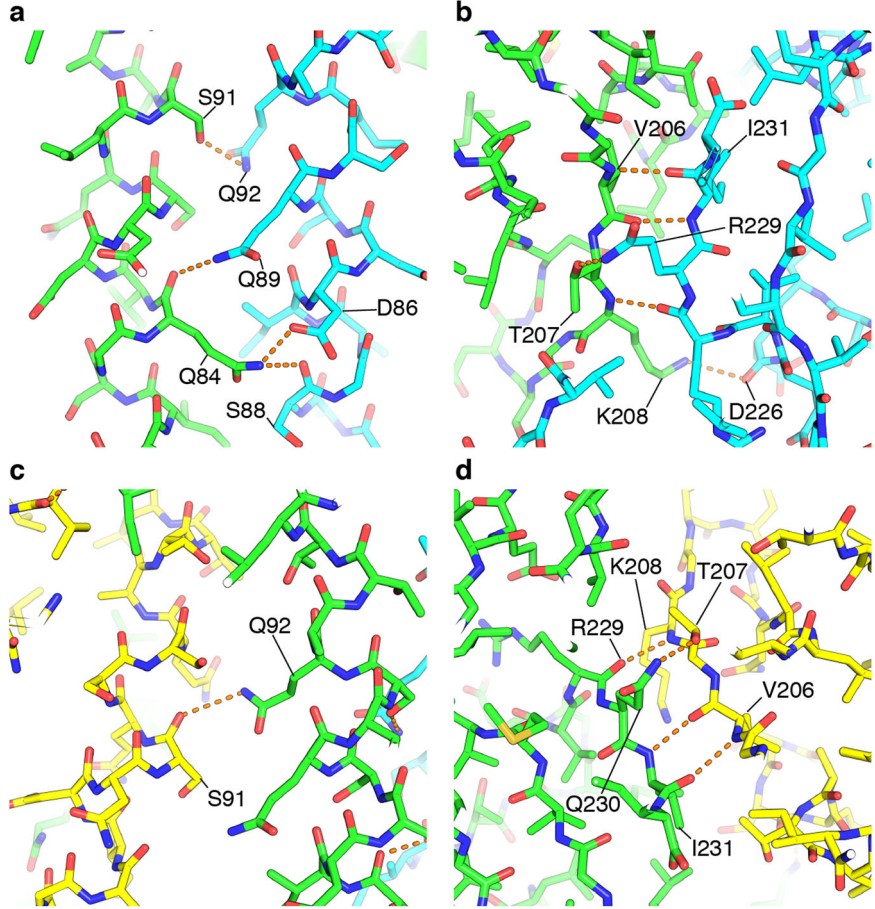

**Fig. 3 Inter-subunit interaction in the hook cap. a–d** Polar contacts were analyzed with *PyMOL* and are displayed in orange dashed lines. The subunits are colored in green (subunit A), cyan (subunit B), and yellow (subunit E). Interactions found between subunit A and B (**a**, **b**) and subunit A and D (**c**, **d**). Oxygen and nitrogen atoms are colored in red and blue, respectively.

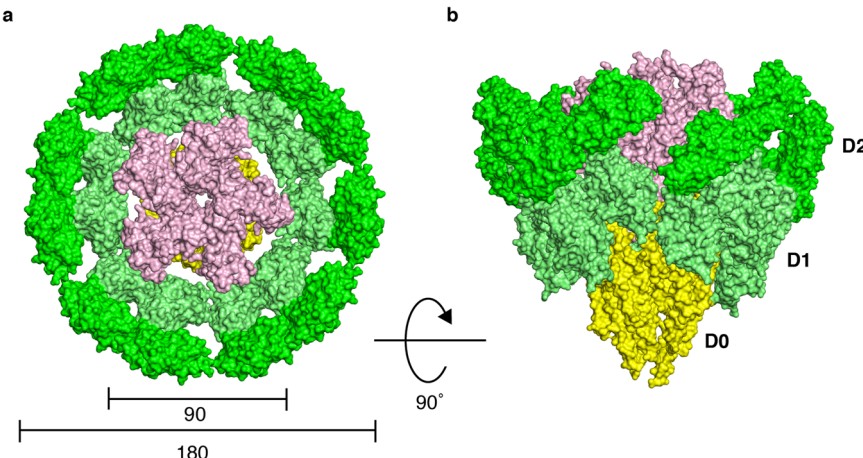

**Fig. 4 A possible docking model of the hook cap on the hook. a, b** The hook cap was fitted into the growing tip of the hook (PDB-id: 3A69). The domains of FlgE (402 residues from *S. enterica*) in the hook are labeled as D0 (1–24, 367–402), D1 (71–144 and 286–357), and D2 (145–285). The hook cap (light pink) and the hook (D0; yellow, D1; lime, D2; green) are shown in solid surface representation and viewed from the top (**a**) and side (**b**). The diameters of the hook cap and the hook are 90 Å and 180 Å, respectively. For clarity, only 11 molecules of FlgE are displayed for the hook.

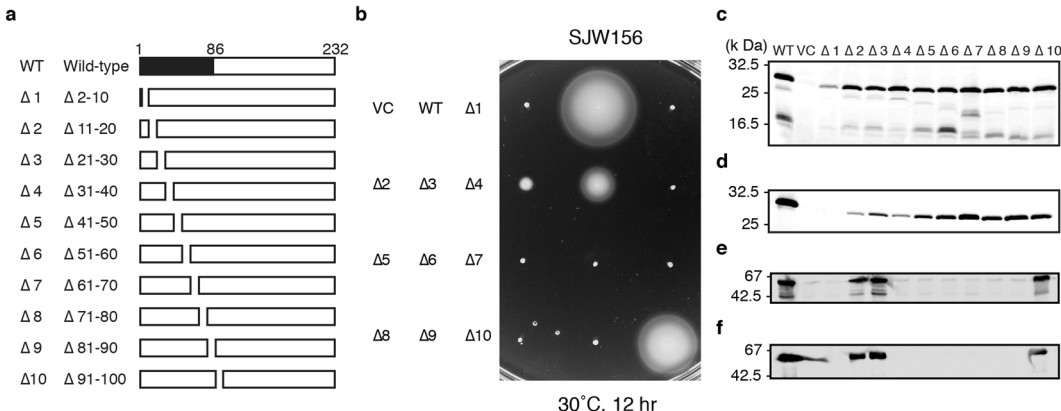

**Fig. 5 In-frame ten-amino-acid deletion analysis of FlgD. a** A series of deletion constructs of the N-terminal domain of FlgD used in this assay. **b** Swarming motility assay of the deletion variants on soft-agar plate. **c, d** Immunoblot analysis of cellular and supernatant fractions of SJW156 complemented by the deletion variants. Protein levels of cellular expression (top) and secretion (bottom) of the FlgD variants (**c**) and FliC (**d**) from *S. enterica* SJW156 complemented by plasmids indicated above the top panel of **c**. "WT" represents wild-type FlgD expressed in SJW156 and "VC" represents a vector control.

were also examined (Fig. 5c, d). The cellular expression levels of FlgD variants were almost equal to that of wild-type FlgD, except Δ1, which showed a markedly reduced expression level. On the other hand, the secretion levels of FlgD variants detected in the supernatant were quite variable. The FlgD variant was not at all secreted by the cells transformed with Δ1, indicating that the first ten residues are responsible for FlgD export by the flagellar type III protein export system. The secretion levels of FlgD variants with Δ2 and Δ4 were significantly lower than those with other variants, suggesting that their motility defect is mainly due to an inefficient assembly of the hook cap. The low secretion level of Δ4 is due to the loss of FlgD secretion signal by the type III protein export[33]. In agreement with the motility restoration by Δ2, Δ3, and Δ10, albeit in variable levels, the expression and secretion of FliC, which forms the filament, were confirmed to be the wild-type levels (Fig. 5d). The reduced motility of the cells with Δ2 and Δ3 is likely due to inefficient hook cap assembly by these FlgD variants. FliC was detected neither in the cellular nor in the supernatant fractions of the cells with FlgD variants Δ1 and Δ4–9, suggesting that the hook cap and, therefore, the hook were not assembled. Thus, the regions of FlgD, which are important for

hook assembly function, were found to be residues 31–90. To analyze whether these FlgD variants could form the pentamer cap as the wild type, native polyacrylamide gel electrophoresis (PAGE) analysis under a non-denatured condition was used to detect oligomer formation. Wild-type and deletion variants of FlgD were expressed in *E. coli* BL21(DE3) cells and the whole-cell fractions were analyzed by immunoblotting using antibody against FlgD (Supplementary Figs. 5 and 8). For wild-type FlgD, the pentamer and monomer were separated by different migration rates. The pentamer was not detected for the cells expressing Δ7, Δ8, and Δ10. The reasons why Δ10 complemented SJW156 to a significant degree even though it did not form the pentamer are unclear. Perhaps, the deletion of residues 91–100 might have caused a rearrangement of the C-terminal domain with its orientation not favorable for pentamer formation. As FlgD is once unstructured and exported by the flagellar type III export system to the distal end of the rod through its narrow central channel to assemble into the hook cap, the distal tip structure of the rod is likely to function as a template for hook cap assembly even for the FlgD variant Δ10, which cannot form the pentamer cap in solution by itself.

## Discussion

We hereby report the first crystal structure of the pentameric hook cap from *S. enterica*. Other experimental data also support that the hook cap is composed of five subunits of FlgD. Unfortunately, some parts of the N-terminal domains including the N terminus were not included in the model due to poor electron density. However, FlgD$_{74-232}$, which lacks most of the N-terminal domain of FlgD, could not form a pentameric complex of the hook cap. The reason why this region is susceptible to limited proteolysis by trypsin in the FlgD monomer become clear from the hook-cap structure. The N-terminal domain of the FlgD monomers are flexible and exposed in solution, and therefore not protected against proteolysis, unlike those of the hook cap pentamer forming a helix bundle. The N-terminal missing part in FlgD$_{74-232}$ completely overlap with the minimum requirement region of FlgD for hook assembly as reported by Kutsukake and Doi[23], although the N-terminal region alone weekly complemented as the hook cap without the C-terminal region of FlgD. Although the N-terminal region of the hook cap undergoes oligomerization, the C-terminal region contributes to the efficient assembly of the hook.

During hook assembly, FlgD partially plugs the central channel of the hook at the tip as a pentameric complex to allow unfolded hook protein FlgE to exit the channel and to be folded and incorporated into the hook by preventing it from diffusing away. The hook cap is kept attached at the distal end of the hook until the hook grows up to a length of about 55 nm[22]. During hook assembly, the ruler protein FliK is occasionally secreted to measure the length of the hook, for which the N-terminal end of FliK must interact with FlgD of the hook cap[34]. In *fliK*-deficient strains, the hook length control is impaired. The hook cap structure alone does not tell us how the N-terminal end of FliK binds to FlgD, whereas the elongated N-terminal chain of FliK measures the length of the hook. However, it is likely that the N-terminal end of FliK interacts with the N-terminal domain of FlgD, which appears to be partially inserted into the central channel of the hook as in the putative model of the hook-cap complex shown in Figs. 4 and 6.

In *S. enterica*, hook assembly requires the FlgD pentamer as the hook cap and filament assembly requires the FliD pentamer as the filament cap. The filament assembly mechanism has been proposed based on the structure of the filament-cap complex where the FliD pentamer with a pentagonal top plate and five leg domains is attached to the concaved end of the filament with the legs through the symmetry mismatch between the pentamer and the half-subunit staggered 11 protofilaments[16,17]. The FliD cap

provides a large enough space underneath the top plate for exported flagellin molecules in unfolded conformation to be folded and incorporated into an appropriate assembly position at the distal end of the filament without diffusing away (Supplementary Fig. 6). Because of this symmetry mismatch, the filament cap is thought to rotate and lifted along the helical structure of the filament to prepare the next assembly site for flagellin molecules that are exported to the distal end one after another. Although the subunit structures of the filament and hook, and their interactions are distinct from each other for their different mechanical properties and functions, because the overall structure of the hook is similar to that of the filament, both consisting of 11 protofilaments, the hook cap was expected to have a similar structure and mechanism with the filament cap. However, FlgD shows poor protein sequence similarity to FliD and their sizes are also remarkably different with 467 amino acid residues for FliD and 232 for FlgD. Now the structure of the hook cap shows a definitive difference from that of the filament cap. The N-terminal chains of five FlgD subunits form one α-helical bundle as the central stalk, which is likely to plug into the central channel of the growing hook, whereas the filament cap provides a space formed by the top plate and five leg domains just above the central channel of the filament as a folding chamber for exported flagellin monomers[16]. Instead, the hook cap appears to provide a space for FlgE folding around the stalk underneath the petal-shaped head (Figs. 4 and 6). However, the same symmetry mismatch seems to be utilized in both structures for promoting hook and filament assembly, suggesting that the hook cap also rotates and is lifted as the filament cap[16] every time a FlgE molecule is exported, folded, and incorporated into the hook. Thus, they seem to share the same mechanism for facilitating helical protein assembly even though their overall shapes are quite different from each other.

Pseudorevertants from hook assembly-deficient FlgE mutant strains of *S. enterica* were isolated[24,35] and the mutation positions were mapped on one surface of FlgD exclusively (Supplementary Fig. 7). It was thought that this surface may be facing the opening of the hook central channel to trap exported FlgE for folding and assembly, but it was not the case. In the hook-cap structure, this surface is located on one side of each petal domain, facing the gap between the two neighboring petal domains (Supplementary Fig. 7). Thus, it remains unclear how the C-terminal petal domain of FlgD contributes to trapping a FlgE molecule and helping its folding and assembly into the hook. The gap between the petal domains may be the site for FlgE trap.

Recently, a high-resolution structure of the hook from *Campylobacter jejuni* was revealed by cryoEM image analysis and it

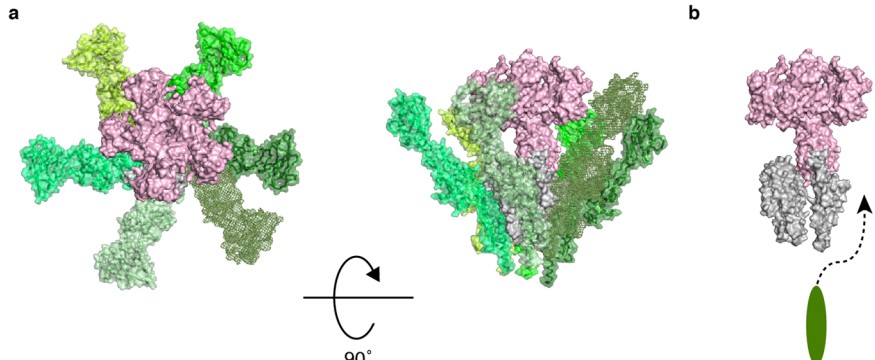

**Fig. 6 A model of hook assembly by the hook cap. a** The pentamer hook cap (light pink) and ten FlgE subunits of the hook (five different greenish colors and gray) are shown in solid surface representation viewed from the top and side. Only domain D0 of FlgE is shown in gray for the first five subunits from the distal end of the hook. The next five FlgE subunits are shown in green, lime, smudge, limon, and pale green. **b** The five subunits in the greenish colors are removed from the right panel of **a** and a newly exported FlgE molecule is added as an elongated ellipse. The N-terminal part of FlgD missing in the model of the hook cap might occupy the space for domain D0 of the FlgE molecule to be incorporated.

showed that the whole structure is made up with 11 protofilaments[36]. The amino acid sequence of *C. jejuni* FlgE is twice longer than that of *S. enterica*. *C. jejuni* FlgD (Cj-FlgD) consists of 294 amino acid residues, which is also larger than the 232-residue FlgD of *S. enterica*, with 23% and 38% sequence identity and similarity, respectively. The extra sequence found in Cj-FlgD might be involved in the interactions with the extra domains found in the *C. jejuni* FlgE in the hook to facilitate hook assembly. To understand the molecular mechanisms of the hook assembly in different bacterial species in more detail, the hook-cap complex structures are greatly desired. Recent advances in cryoEM image analysis should allow us to solve the structures of such hetero-multiprotein complexes.

## Methods

**Protein expression and purification of *S. enterica* FlgD.** The *flgD* gene was PCR-amplified from the genomic DNA of *S. enterica* serovar Typhimurium SJW1103 using primer 1, 5′-GTAAAGGAGGCGCATATGTCTATTGCC-3′ (NdeI, underlined) and primer 2, 5′-CTGGATCCTGATAAGTGTAAGGGCTTAG-3′ (BamHI, underlined), with restriction enzyme sites. The full-length *flgD* gene was cloned into a pET3c vector (Novagen) to generate pHMK1901. The amplified DNA fragment was confirmed by using DNA Sequencer ABI 377 with BigDye Terminator Cycle Sequence Kit (Thermo Fisher Scientific). This plasmid was used to express protein in *E. coli* BL21(DE3)pLysS (Novagen). Cells were grown at 37 °C in Luria-Bertani (LB) media supplemented with 50 mg L$^{-1}$ ampicillin and 30 mg L$^{-1}$ chloramphenicol overnight, and diluted 100 times into fresh LB medium. Cells were grown at 37 °C until cell density reached 0.5–0.6 at 600 nm. IPTG (isopropyl β-D-thiogalactopyranoside) was added to the culture medium, to a final concentration of 1 mM for induction of the protein. After an additional 4 h culture, the cells were collected by centrifugation at 4000 × $g$ for 10 min and washed with 0.85% (w/v) sodium chloride prior to storage at −80 °C. The frozen cells were suspended in 20 mM Tris-HCl pH 8.0, 1 mM EDTA (ethylenediaminetetraacetic acid) (buffer A) containing cOmplete protease inhibitor cocktail (Roche). The cells were lysed by sonication and centrifuged at 105,000 × $g$ for 30 min, to remove undisrupted cells and insoluble materials. The supernatant fraction containing FlgD was applied to a Q-Sepharose HP column (GE Healthcare) equilibrated in buffer A. After unbound species were washed thoroughly, FlgD was eluted by a liner-gradient of sodium chloride. The fractions containing FlgD were precipitated by ammonium sulfate and re-suspended in 20 mM Bis-Tris-HCl pH 6.0, 1 mM EDTA, 0.4 M ammonium sulfate (buffer B). The protein was loaded onto a Butyl-Sepharose column (GE Healthcare) equilibrated in buffer B. After unbound species were washed thoroughly, FlgD was eluted by a liner-gradient of ammonium sulfate. The fractions containing FlgD were precipitated by ammonium sulfate. The suspension was kept for 1 h on ice before spinning down and re-suspended in a minimum volume of 20 mM Bis-Tris-HCl pH 6.0, 0.3 M sodium chloride (buffer C). After removing insoluble materials by centrifugation, the protein was applied onto Superdex 200 HR16/60 (GE Healthcare) pre-equilibrated in buffer C. The fractions containing FlgD were pooled and concentrated. FlgD was stored at −20 °C until use.

**Protein biochemical characterization.** Limited proteolysis and blue-native PAGE (BN-PAGE) analysis were carried out. After the pentameric and monomeric fractions of FlgD were separated by gel filtration, proteins were subjected to limited proteolysis by trypsin for times from 1 to 120 min at 25 °C. Each reaction was stopped by adding TPCK (N-$p$-Tosyl-L-phenylalanine chloromethyl ketone) and heating at 95 °C for 5 min in a sample buffer containing sodium dodecyl sulfate (SDS). Sample preparation for BN-PAGE was followed by the manufacturer's protocol. Chemical cross-linking experiment was followed by a protocol described previously[15] with slight modifications. The purified FlgD protein was chemically modified by cross-linkers EDC (1-ethyl-3-(3-dimethylaminopropyl) carbodiimide) with *sulfo*-NHS (hydroxy-2,5-dioxopyrrolidine-3-sulfonicacid) or DTSSP (3,3′-dithiobis(sulfosuccinimidyl propionate)).

**Purification, crystallization, and structural determination of a fragment FlgD$_{74-232}$.** A plasmid for expression of the fragment from Leu-74 to the end of *S. enterica* FlgD was constructed by PCR using primer 3, 5′-GGAATTCC-TATGCTGAATACGACGCTGGGG-3′ (NdeI, underlined) and primer 2. The DNA fragment was cloned into a vector plasmid pET14b (Novagen) to create pHMK2906, which produces FlgD$_{74-232}$ fused with a hexa-histidine tag and an enterokinase digestion site derived from the plasmid in its N terminus. FlgD$_{74-232}$ was purified from the soluble fraction after disruption of *E. coli* BL21(DE3) cells harboring pHMK2906. The protein was purified by nickel-nitrilotriacetic acid affinity chromatography and followed by gel-filtration chromatography. The protein was concentrated to 60–70 mg ml$^{-1}$. Initial crystallization screenings of FlgD$_{74-232}$ were carried out using crystallization kits Wizard I, Wizard II, and Wizard III (Emerald Biosciences), and CrystalScreen I and II (Hampton Research). Drops were prepared by mixing 2 μl of the protein solution with an equal amount of the reservoir solution and were equilibrated against 0.1 ml of the reservoir solution by the sitting-drop vapor-diffusion technique at three different tempera-

tures (278, 288, or 293 K). Initial crystals of FlgD$_{74-232}$ were obtained by 0.2 M ammonium citrate dibasic, 30% (w/v) polyethylene glycol (PEG) 3350 at 293 K. After extensive screening of PEG solution, diffraction quality crystals grew from a solution containing 0.2 M ammonium citrate dibasic, 24% (w/v) PEG 2000 at 293 K. Heavy-atom derivative crystals were prepared by soaking crystals in the solution containing potassium hexachloroosmate (IV) (K$_2$OsCl$_6$) for 8–12 h. Crystals were cryo-protected in a solution containing 90% (v/v) reservoir solution and 10% (v/v) ethylene glycol for a few seconds prior to freezing in liquid nitrogen. All diffraction data were collected on a charge-coupled detector (CCD) of SPring-8 (Harima, Japan). The diffraction data were indexed and integrated with *MOSFLM*[37] and scaled with *SCALA* from the *CCP4* program suite[38]. Molecular replacement trials by *MOLREP*[39] or *Phaser*[40] using search models from the deposited structures were unsuccessful. Eventually, initial phase for the Os-derivative crystal was calculated with *auto.sol* module in the *PHENIX* suite[41]. Phase was extended to a resolution of the native crystal and the initial model was built with *AutoBuild* module in the *PHENIX* suite. After manual modification of the model with *Coot*[42], the structure was iteratively refined with the *phenix.refine*[43] module in the *PHENIX* suite. The final model of FlgD$_{74-232}$ was validated with *Molprobity*[44]. The statistics of data collection and structural refinement are summarized in Table 1.

**Crystallization and structural determination of FlgD.** Initial crystallization screening was performed by hanging-drop vapor-diffusion method with commercially available crystallization kits. Equal volumes of screening solutions (2 μl) were mixed with the protein solution. Small plate-like crystals appeared within 2 or 3 days at 16 °C using the No. 40 solution of 0.1 M sodium citrate (pH 5.6), 30% (w/v) PEG 4000, and 30% (v/v) 2-propanol in CrystalScreen (Hampton Research). After refinement, the initial condition, diffraction quality crystals were finally obtained by mixing with a reservoir solution containing 16% (w/v) PEG 2000, 12% (v/v) 2,3-butanediol, 8% (v/v) glycerol, and 0.1 M sodium citrate (pH 5.6). To reduce evaporation rate under hanging-drop vapor-diffusion method, 0.4 ml silicon oil was overlaid onto 1 ml of the reservoir solution in Linbro 24-well crystallization plates (Hampton Research). Micro-seeding technique was essential for conducting optimal crystal growth of the crystals. Crystals were grown to the maximum size within a couple of weeks at 277 K. In *E. coli* B834(DE3) (Novagen), seleno-methionine (Se-Met)-substituted FlgD was expressed[45] and purified as described above for the native protein. Se-Met FlgD crystallized in the same condition as the native protein. Crystals were frozen in nylon loops (Hampton Research) by plunging into liquid nitrogen. A multiple-wavelength anomalous dispersive data set was collected under a cryogenic temperature of 100 K from single crystals at beamline BL41XU at SPring-8 (Harima, Japan). The native data set of the FlgD crystal was collected at 40 K in helium gas flow. All diffraction data collected on a CCD were processed by using *MOSFLM* and scaled with *SCALA* from *CCP4* program suite. Initial phase was calculated with *SHARP/autoSHARP*[46]. The structural model of FlgD$_{74-232}$ was fitted into the initial electron density with *MOLREP*. After manually building an initial model of FlgD with *Coot*, the structure was iteratively refined with *Refmac5*[47] and *phenix.refine* imposing secondary structure and non-crystallographic symmetry restrains. The final model of the hook cap was validated with MolProbity. The statistics of data collection and structural refinement are summarized in Table 2.

**Graphic preparation.** All structural figures were prepared with *PyMOL* (http://www.pymol.org) or *UCSF Chimera*[48].

**Molecular genetic assay.** For protein secretion assay, *S. enterica* cells were grown at 30 °C until the cell density reached 1.0–1.2 at an absorbance of 600 nm. Cells were spun down and culture supernatant fractions were collected separately. Proteins were precipitated by adding trichloroacetic acid at a final concentration of 10% (v/v) and spun down by centrifugation. Pellets were completely suspended in 1 M unbuffered Tris-base before mixing with Tris/SDS gel loading buffer. Proteins were detected by immuno-blotting with polyclonal anti-FlgD antibody as described elsewhere.

For motility assay, *S. enterica* cells were transformed by electroporation using *E. coli* Pulser (Biorad) with plasmids. Fresh transformants were selected on an LB plate containing 100 μg ml$^{-1}$ of ampicillin and inoculated onto 0.4% (w/v) soft-tryptone agar plate containing 100 μg ml$^{-1}$ of ampicillin and incubated at 30 °C for 5–6 h.

**Reporting summary.** Further information on research design is available in the Nature Research Reporting Summary linked to this article.

## Data availability

Coordinates and structure factors have been deposited in the Protein Data Bank (PDB) under accession codes 7EH9 (monomeric form of FlgD) and 7EHA (pentameric form of FlgD).

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

## Acknowledgements

We thank beamline staffs at Spring-8 (Harima, Japan) for technical support in the use of BL41XU (proposal number 2008B1382) and BL44XU (proposal numbers 2015B6501, 2016A6601, and 2016B6601). We thank Yukio Furukawa and Midori Yamane from Osaka University for help with protein characterization, and Tohru Minamino for critical reading of the manuscript. This work was supported in part by JSPS KAKENHI Grant Numbers 17K07318 and 20K06581 to H.M., 15H02386 to K.I., JP25000013 to K.N., MEXT KAKENHI Grant Number 23115008 to K.I., and by a direct funding from OIST to F.A.S.

## Author contributions

H.M. designed the project and performed the experiments. K.I., Y.-H.Y. and F.A.S. helped in data collection and structure determination. H.M. prepared the figures. F.A.S. and K.N. supervised the project. H.M., F.A.S. and K.N. wrote the manuscript with the help of the other authors.

## Competing interests

The authors declare no competing interests.
