## [Transparent Peer Review File · Communications Biology]

Reviewers' comments:

Reviewer #1 (Remarks to the Author):

The paper "Structure of the bacterial flagellar hook cap provides an insight into hook assembly mechanism" by Matsunami et al. describes the crystal structure of the pentameric assembly of FlgD from *Salmonella*. The structure of the C-terminal domain of FlgD from other bacterial species is already known, but this is the first time that the structure of the entire protein is presented. This is particularly relevant, since the N-terminal domain is responsible for the pentamerization of the structure. The assembly allows the authors to propose a quite convincing model for the hook-capping assembly. Also the discussion about the mechanism of elongation of the hook is interesting, despite highly speculative.

Minor points:

At page 7, the authors state that the previously solved FlgD structures from other bacteria share a conserved structure for the C-terminal domain with *Salmonella* FlgD. It should be interesting if they could speculate if the proposed mechanism of hook-capping for *Salmonella* could be valid also for the other bacteria.

In conclusion, the paper is well written and the structures properly done, despite the limited resolution of the pentameric model. It is worth being published.

Reviewer #2 (Remarks to the Author):

This manuscript submitted by Matsunami et al. applied x-ray crystallography and molecular genetic approaches to resolve the structure of hook cap protein FlgD and define the critical regions for the secretion of flagellar hook in *Salmonella enterica*. The authors reported two crystal structures of FlgD; one contains residues 55-232 and another contains residues 74-232. The extra 20 amino acid residues in the FlgD(55-232) forms a long helix which interacts with its neighboring FlgD molecule for the pentamer formation. The structure of remaining core of FlgD (74-232) resembles those resolved in *Xanthomonas*, *Pseudomonas* and *Helicobacter pylori*. The authors further docked the model to the cryo-EM structure of the hook. In addition, the authors carried out *in vivo* complementation studies using different deletion mutants to map the regions essential for motility, expression and secretion of FlgD.

There are some major concerns that need further clarification and supplemental data.

1. While this manuscript describes the importance of the N-terminal helix for oligomerization, the core domain of the structural model of FlgD share high structural similarity with orthologues in three other bacteria (PDB ID: 4ZZK, 3OSV, 3C12) published before. The novelty of the findings presented in the manuscript is one of the major concerns of the reviewer.
2. Quality of FlgD (55-232) structure. Statistics of x-ray diffraction data (Supp Table2). A very high Rmerge value (73.7%) at outer resolution shell (3.1Å). The authors should reduce the resolution cutoff to 3.2Å or 3.3Å. The PDB validation report also indicate very high RSRZ outliers. The authors should inspect the fitness between the structural model the data carefully.
3. Some suggestions to strengthen the biological relevance of the pentameric FlgD observed in the crystal form. *In vivo* cross-linking to confirm the N-terminal helical region with hook proteins. EM study to examine the hook formation.

Some minor points are listed below.

1. Figure 1 c. What are the two numbers in the bracket next to P and M? If there are the corresponding molecular size of P and M, why it is smaller than the size predicted from the elution profile of gel filtration?

2. Line 90, and Figure 1d and e. No molecular size marker to reference the oligomers are pentamer and decamer.
3. Line 175-179. Lack of details of how the current structural model can provide implications for the hook assembly.
4. Figure 5d upper panel. Why the expression of FliC is blocked in mutants 1, 4-9?

To Reviewer #1

> The paper “Structure of the bacterial flagellar hook cap provides an insight into hook assembly mechanism” by Matsunami et al. describes the crystal structure of the pentameric assembly of FlgD from Salmonella. The structure of the C-terminal domain of FlgD from other bacterial species is already known, but this is the first time that the structure of the entire protein is presented. This is particularly relevant, since the N-terminal domain is responsible for the pentamerization of the structure. The assembly allows the authors to propose a quite convincing model for the hook-capping assembly. Also the discussion about the mechanism of elongation of the hook is interesting, despite highly speculative.

Thank you for your supportive comments. The discussion may be somewhat speculative, the quality of the structure is high enough to show the overall shape of the functional hook cap, so we can suggest that the assembly process of the hook proceeds in a quite different manner from that of the filament of the bacterial flagellum.

> Minor points:

> At page 7, the authors state that the previously solved FlgD structures from other bacteria share a conserved structure for the C-terminal domain with Salmonella FlgD. It should be interesting if they could speculate if the proposed mechanism of hook-capping for Salmonella could be valid also for the other bacteria.

Thank you for the suggestion. In bacteria, FlgD shares similar domain organization of the N-terminal region essential for hook assembly and the C-terminal regions for enhancing the hook export [Moriya et al (2013) doi: 10.2142/biophysics.9.63].

> In conclusion, the paper is well written and the structures properly done, despite the limited resolution of the pentameric model. It is worth being published.

Thank you for your positive comment.

To Reviewer #2

> This manuscript submitted by Matsunami et al. applied x-ray crystallography and molecular genetic approaches to resolve the structure of hook cap protein FlgD and define the critical

regions for the secretion of flagellar hook in *Salmonella enterica*. The authors reported two crystal structures of FlgD; one contains residues 55-232 and another contains residues 74-232. The extra 20 amino acid residues in the FlgD(55-232) forms a long helix which interacts with its neighboring FlgD molecule for the pentamer formation. The structure of remaining core of FlgD (74-232) resembles those resolved in *Xanthomonas*, *Pseudomonas* and *Helicobacter pylori*. The authors further docked the model to the cryo-EM structure of the hook. In addition, the authors carried out in vivo complementation studies using different deletion mutants to map the regions essential for motility, expression and secretion of FlgD.

>

> There are some major concerns that need further clarification and supplemental data.

> 1. While this manuscript describes the importance of the N-terminal helix for oligomerization, the core domain of the structural model of FlgD share high structural similarity with orthologues in three other bacteria (PDB ID: 4ZZK, 3OSV, 3C12) published before. The novelty of the findings presented in the manuscript is one of the major concerns of the reviewer.

We have shown here the first crystal structure of the pentameric form of FlgD, which is the functional form of FlgD in its hook-capping function. The structure provides the protein-protein interactions for pentamer formation through the N-terminal region. Although the structure of the C-terminal domain of FlgD shares a similarity with other FlgD proteins but none of them could ever propose a hook cap model due to the truncation of the N-terminal region for crystallization. We provided supportive evidence on the importance of the N-terminal region by deletion analysis and thereby proposed that this pentameric form of FlgD acts as the hook cap for hook assembly.

> 2. Quality of FlgD (55-232) structure. Statistics of x-ray diffraction data (Supp Table2). A very high Rmerge value (73.7%) at outer resolution shell (3.1Å). The authors should reduce the resolution cutoff to 3.2Å or 3.3Å. The PDB validation report also indicate very high RSRZ outliers. The authors should inspect the fitness between the structural model the data carefully.

Thank you for the suggestion. For the pentamer structure, we reprocessed the diffraction data to 3.3 Å and obtained significant improvement of the data statistics. The R-merge value markedly decreased in the highest resolution shell (40.1%) and the RSRZ values were also improved to 7.0% (originally 9.0% at 3.1 Å). We also reprocessed the diffraction data of the monomeric form of FlgD to 2.2 Å, and the statistics were improved to 6.7% (originally 15.9% at 2.1 Å).

> 3. Some suggestions to strengthen the biological relevance of the pentameric FlgD observed in the crystal form. In vivo cross-linking to confirm the N-terminal helical region with hook proteins. EM study to examine the hook formation.

Thank you for the suggestion. Fumiaki Makino and his colleagues at Osaka University have carried out cryoEM analysis of the hook cap structure at the tip of the hook-basal body and successfully visualized the pentameric complex of FlgD there. We now refer to this study in the revised manuscript as “Makino et al, personal communication”.

>

> Some minor points are listed below.

> 1. Figure 1 c. What are the two numbers in the bracket next to P and M? If there are the corresponding molecular size of P and M, why it is smaller than the size predicted from the elution profile of gel filtration?

These values indicate the molecular masses of the pentamer (P) and monomer (M) calculated from the amino-acid sequence. The molecular shape of FlgD is not spherical but elongated, and therefore the apparent molecular size of the protein estimated by gel-filtration or non-native gel electrophoresis are larger than expected.

> 2. Line 90, and Figure 1d and e. No molecular size marker to reference the oligomers are pentamer and decamer.

Molecular size markers of cross-linked haemoglobin (32K, 48K, 64K) and cross-linked albumin (132K, 198K) are added on the left of Figure 1d and e.

> 3. Line 175-179. Lack of details of how the current structural model can provide implications for the hook assembly.

We discussed the implication of the current structural model for the hook assembly in detail in Discussion of the original manuscript.

> 4. Figure 5d upper panel. Why the expression of FliC is blocked in mutants 1, 4-9?

In *Salmonella* cells with these FlgD deletion mutants, the hook is not formed, and therefore

the substrate specificity of the flagellar type III protein export apparatus does not switch from the rod-hook type to the filament type, so that the late flagellar genes including *fliC* are not transcribed. [Kutsukake, K. et al (1990) J Bacteriol **174**, p.741-747].

REVIEWERS' COMMENTS:

Reviewer #1 (Remarks to the Author):

In the present form the paper in my opinion is worth of being published.

Reviewer #2 (Remarks to the Author):

The authors have responded the comments from both reviewers. I recommend the acceptance of the revised manuscript after some editing of English usage.